# Influential node identification method based on multi-order neighbors and exclusive neighborhood

Feifei Wang[1], Zejun Sun[1]*, Guan Wang[1], Bohan Sun[2]

**1** School of Information Engineering, Pingdingshan University, Pingdingshan, Henan, China, **2** School of Mechanical and Power Engineering, Zhengzhou University, Zhengzhou, Henan, China

* szj@pdsu.edu.cn

## Abstract

In a complex network, the identification of node influence and the localization of key nodes play a crucial role in analyzing network structure and determining the positioning of nodes for information transmission control, resource redistribution, and network regulation. In this study, we propose a method for identifying influential nodes called "Multi-order Neighbors and Exclusive Neighborhood" (MNEN) after analyzing and investigating existing methods in the field. The MNEN method calculates a node's influence based on two factors: the node itself, its neighboring nodes, and its exclusive neighborhood. The influence of the node itself is determined by its degree value and K-shell ($Ks$) value, while the influence contribution of the neighbor node is calculated based on its degree value, $Ks$ value, and the contribution from its exclusive neighbor node. To evaluate the algorithm's performance, we employ the SIR model as the benchmark and conduct simulation experiments to validate the MNEN method, comparing the results with other influential node identification methods. Our analysis demonstrates that the algorithm accurately identifies influential nodes in networks of different scales, yielding a positive overall impact and demonstrating a certain level of universality.

## Introduction

In recent years, scholars have paid considerable attention to the study of complex networks [1–4], and many real-world relationships and problems can be represented as networks [5–8], such as biological networks [9], power networks [10] epidemic prevention and control networks [11], etc. As needed, the abstracted networks can be studied to find solutions to various problems. Among them, identifying influential nodes is one of the current research hotspots [12–15]. In complex networks, the status of different nodes varies, and the importance of nodes with significant influence is greater than that of ordinary nodes; identifying these nodes helps to comprehend the

**Data availability statement:** All relevant data and source code are fully available without restriction at the GitHub repository: https://github.com/wffei123/MNEN.

**Funding:** This work was supported by the Key Scientific Research Projects of Colleges and Universities in Henan Province of China under Grant 25A520040 (to ZS), and the Science and Technology Research Project of the Science and Technology Department in Henan Province of China under Grant 252102210028 (to ZS). The funders had no role in study design, data collection and analysis, decision to publish, or preparation of the manuscript.

**Competing interests:** The authors have declared that no competing interests exist.

network structure at a deeper level, identify the network's key hubs, control information dissemination, optimize resource allocation, and many others. In criminal gangs, for instance, it is possible to identify the backbone members and ordinary members [16,17]. In disease transmission networks, it is possible to identify people or objects critical to infection [18–20]; in social networks, it is possible to identify the most authoritative key people; in public opinion, the most valuable information disseminators can be sought [21–23]; in logistics networks, key hubs can be identified [24]; etc. Furthermore, the application of node identification and complex network analysis extends beyond single disciplines, encompassing interdisciplinary contexts like biology (for instance, the analysis of protein-protein interaction networks), neuroscience (such as the examination of brain connectivity), and environmental science (including the study of ecosystem networks).

In the process of identifying the influential nodes of complex networks, many methods have emerged. The early classics include Closeness Centrality (CC) [25], Betweenness Centrality (BC) [26], PageRank (PR) [27], K-shell [28], etc. More recent methods include GSM [29], ProfitLeader (PL) [30], Gravity [31], GIN [32], KBKNR [33], and ELKSS [34]. Different methods consider different factors when identifying node influence. Typically, global or local thinking is adopted. At the same time, neighbor node, the location of nodes, and the contribution made by nodes' neighbors are introduced. Each method has its own advantages for calculating node influence, as well as its own disadvantages in terms of time complexity, effect on recognition, etc.

Based on the existing influential node identification methods, this paper presents the influential node identification method MNEN, which takes into account the node's degree, its position, and the exclusive neighborhood of neighbor and neighbor nodes with respect to the node. This method includes two factors when calculating the influence of a node: the node itself and its neighbors. Calculate the influence provided by the node itself, based on its degree and *Ks* value. The influence provided by neighbor nodes consists of two components: neighbor nodes and the node's exclusive neighborhood of neighbor nodes. The basic ideas and major contributions of this approach are described below.

### Basic idea

According to the MNEN method, the influence of a node is attributed to two factors: the node itself and its neighbors, or the contribution of neighbor nodes to it.

When a node has strong influence and its neighbors can provide additional influence, the node's significance in the network increases. Moreover, the location of nodes is also extremely important. The greater the influence of a node, the closer it is to the core. The further a node is from the core, the less influence it has. Therefore, in the MNEN method, the number of first-order neighbors and the location of nodes are considered when calculating the initial importance of each node. Calculate the initial contribution of a neighboring node to this node before calculating the contribution of that node. Here, we present the number of neighbors within one hop of the neighbor node, the *Ks* value, and the neighbor node's exclusive domain with respect to the node. Then, calculate the probability that the neighboring node contributes.

Then, multiply the initial influence of this neighbor node by the probability of obtaining the neighbor node's final influence contribution.

To more clearly explain the basic idea of the MNEN method, it is described here according to the simple network model. Fig 1 is a generating network. In Fig 1, node $v_4$ serves as an example of the fundamental concept.

The influence of $v_4$ is determined jointly by $v_4$ and its neighboring nodes. $v_5$ is one of the neighbor nodes of $v_4$. Consider $v_5$ as an example to calculate and describe its contribution to $v_4$. See Fig 2 for details.

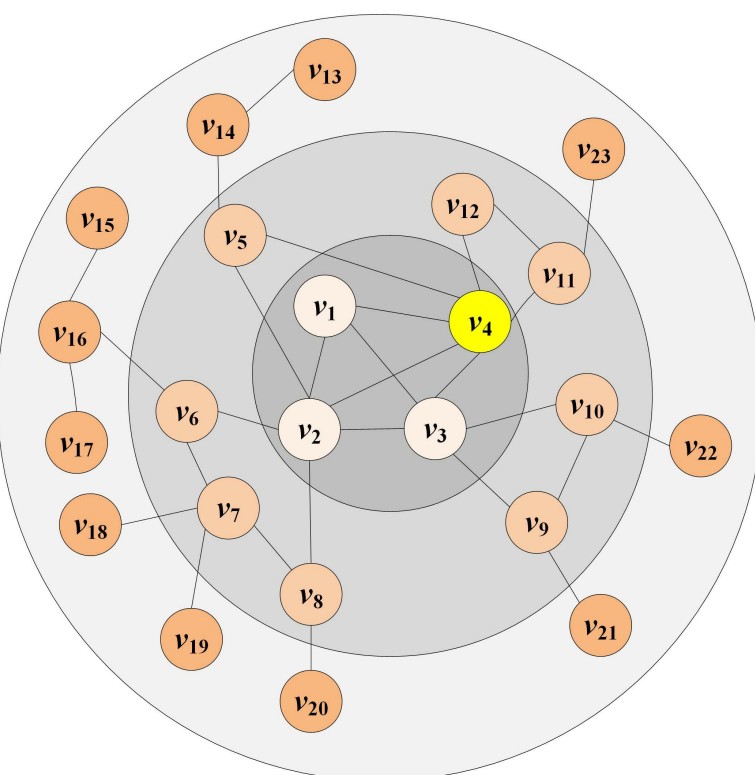

**Fig 1. An example network.**

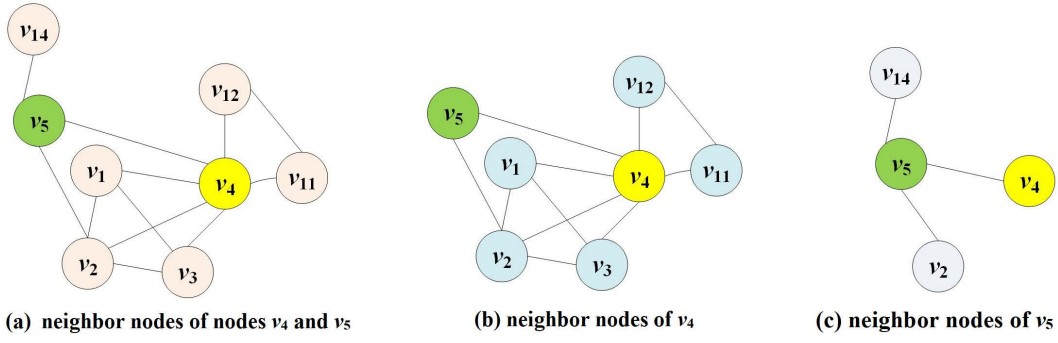

(a) neighbor nodes of nodes $v_4$ and $v_5$ (b) neighbor nodes of $v_4$ (c) neighbor nodes of $v_5$

**Fig 2. Schematic diagram of node $v_4$ as an example.**

Fig 2a depicts nodes $v_4$ and $v_5$, as well as their neighbors. Fig 2b shows $v_4$ and its neighbor node set $\{v_1, v_2, v_3, v_5, v_{11}, v_{12}\}$, and Fig 2c depicts $v_5$ and its neighbor node set $\{v_2, v_4, v_{14}\}$. Where $v_2$ and $v_{14}$ are the common neighbor nodes of $v_4$ and $v_5$, $v_2$ has been included in the calculation of $v_4$'s neighbor importance because it is already a neighbor of $v_4$. Therefore, when calculating the influence of $v_5$ on $v_4$'s contribution, it will be removed to avoid recalculating the influence of $v_2$'s common neighbor. Otherwise, the proportion of $v_2$ in $v_4$'s overall influence will increase, leading to biased results. When calculating the influence provided by $v_5$ to $v_4$, only node $v_{14}$, which is a two-hop neighbor node of node $v_4$ passing through node $v_5$ under certain conditions, i.e., an exclusive neighbor node, is considered. In the example in Fig 2, in order to avoid special cases in which there is no node in the exclusive neighborhood, $v_4$ is included; in comparison to $v_4$, the exclusive neighborhood of $v_5$ is $\{v_{14}, v_4\}$.

## Contributions

MNEN method provides an important idea for identifying key nodes. Its important features are as follows:

(i) A new method for identifying key nodes is presented. In this method, the node's degree and *Ks* value, as well as its location, are utilized. When calculating the contribution provided by neighboring nodes, an exclusive neighborhood is implemented to prevent repeated calculation of the influence of common neighboring nodes.

(ii) The identification result is stable. The method does not involve the setting of parameters, and the identification of influential nodes is not dependent on the adjustment of parameters, resulting in stable results.

(iii) The calculation is simple and the performance is better. Compared with the classical and recent several influential node detection methods, this method is simple to calculate and has a superior detection effect.

The remaining sections of this paper consist of four sections. In the "Related Works" section, we delve into and meticulously examine both the classical and more recent, influential node identification methods. Moving on, in the "The Method of MNEN" section, we elaborate on the MNEN method, focusing primarily on its underlying design concept and the meticulous calculation procedure for determining the node influence value. In the "Experimental evaluations" section, we have elaborated upon the simulation experiment, introduced the datasets for comparison purposes, contrasted MNEN with various methodologies, and delved into the results. Ultimately, we present a concise summation of the proposed techniques, highlighting the areas that require further exploration in subsequent stages.

## Related works

### Classical influential node identification methods

CC [25] considers the distance between nodes from a global perspective, and calculates node influence based on the average distance between nodes. However, this method must calculate the distance between nodes, which has a lengthy calculation time and a high level of complexity, and is best suited for smaller to medium-sized networks. In the BC [26], first, calculate the shortest paths between every pair of nodes in the network, and then determine which node has the greatest number of shortest paths. The significance of a node is proportional to the number of shortest paths that pass through it, which indicates that it is connected to multiple propagation links. However, due to additional calculations, the complexity of this method is high. The PR method [27] obtains the PR value of a web page through multiple iterations. During the calculation process, the new PR value is calculated based on the PR value of the page that links to it and the number of links out of the web page. However, in this method, the PR value of the web page is evenly distributed without considering the influence of the targeted web page, and the average value tends to lead to inaccurate results. Starting from the node with an initial degree of 1 in the network, the K-shell method [28] divides the nodes into multiple layers. This method's calculation is relatively simple, but it only divides nodes into multiple layers. The low degree of influence differentiation between nodes is generally applicable to the initial division of nodes in large networks.

## Recent influential node identification methods

Many scholars have proposed new influential node identification methods based on research into complex network structures and classical methods. Only a few methods are analyzed and described in this section.

In 2021, Aman Ullah et al. proposed the global information method GSM [29]. This method includes two parts when calculating the influence of a node, the self-influence, and the global influence. The self-influence is calculated based on the $Ks$ value of the node, which is the number of nodes. The global influence is equal to the sum of the $Ks$ values of all other nodes and the shortest path ratio. The overall influence of a node is the multiplication of its own influence and the global influence. As a result of the incorporation of global factors, the calculation results are more precise, but the level of complexity is increased. In 2019, Zhongjing Yu et al. proposed a method called PL [30]. The profitability of a node is the product of the node's available resources and the probability of sharing. The available resources of the node equal the sum of the node's degree and the exclusive node's degree. The sharing probability of the node is the Jaccord coefficient between the two nodes. This method determines the importance of nodes by calculating the profitability that each node can provide to other nodes. However, when calculating the node's available resources, it only considers the node's degree and its exclusive neighbor node, which has certain limitations. Lingling Ma et al. proposed the Gravity method [31]. This method determines the influence of a node based on its $Ks$ value and the distance between it and another node. The greater the product of the $Ks$ values of a node and its neighbors (which can be first-order neighbors or multi-order neighbors) and the shorter the shortest distance between the two nodes, the more significant the node. The complexity of this method is higher. Jie Zhao et al. proposed a method based on a global perspective, GIN [32], in which the importance of a node consists of its own importance and global importance. The self-importance is calculated by the degree of the node and the total number of nodes in the entire network, and the natural logarithm is introduced. The greater the degree and proximity of a node, the greater its global significance. During calculation, the method sets a parameter for both parts. While the ratio of self-importance to global importance can be adjusted to meet specific requirements, the parameter setting adds uncertainty to the method identification result. Lixia Xie et al. proposed a method for identifying influential nodes [33]. The method begins by layering the network to determine the $Ks$ value of each node, and then refines the influence of nodes using synthesis degree. The synthesis degree is calculated based on the first-order neighbor nodes and secondary neighbor nodes of nodes, with the influence coefficient adjusting the proportion of the two in the synthesis degree. This method is relatively simple to calculate, but it uses the $Ks$ value to divide the first layer of node influence, which has certain limitations. Furthermore, it does not account for isolated nodes during the calculation process, so it cannot be applied to networks with isolated nodes. Fan Yang et al. proposed the ELKSS method [34], which is based on the extended local sum of $Ks$. This method calculates the influence of a node by first calculating the sum of $Ks$ values (LKSS) of nodes within two hops of each neighbor node, and then adding the LKSS values of all neighbor nodes. This method is simple to calculate, but it only considers the $Ks$ value, which is inadequate for solving the coarse-graining issue.

In addition, there are many other node detection methods, each of which has its own significant factors and benefits, as well as its own drawbacks.

## The method of MNEN

In this section, the MNEN-proposed method is described in detail. First, the concepts and formulas used in the method are described. Next, the method flow is outlined, and the method's performance is validated by analyzing a model network. The last section analyzes the time complexity of the MNEN method.

## Method-related concepts and preliminaries

The significance of nodes must be determined by calculation. In this method, a node's importance is determined by two factors: the node itself and all of its neighbors. Assume that $G=(V, E)$ is an undirected and unweighted network.

**Definition 1** (node's own influence): The own influence of node $v_i$ is determined by the node's $Ks$ value $Ks(v_i)$ and the node's degree $d(v_i)$. The calculation formula is as follows:

$$SI(v_i) = K_s(v_i) + d(v_i) \qquad (1)$$

**Definition 2** (exclusive neighborhood of node): Take node $v_i$ as an example, node $v_j$ is the neighbor node of node $v_i$, $Nei(v_i)$ is the neighbor node set of $v_i$, $Nei(v_j)$ is the neighbor node set of $v_j$, $CN(v_i, v_j)$ is the common neighbor node set of node $v_i$ and $v_j$, and $EN(v_i, v_j)$ is the exclusive neighbor node set of node $v_j$ relative to node $v_i$, i.e., exclusive neighborhood. The specific calculation is as follows:

$$CN(v_i, v_j) = Nei(v_i) \cap Nei(v_j) \qquad (2)$$

$$EN(v_i, v_j) = Nei(v_j) - CN(v_i, v_j) \qquad (3)$$

**Definition 3** (initial influence of neighbor nodes): For node $v_i$, the initial influence provided by its neighbor node $v_j$ is $NI(v_j)$. $NI(v_j)$ consists of two parts: the influence $\Gamma(v_j)$ provided by the node $v_j$ itself for $v_i$ and the influence $I(v_j)$ provided by the exclusive neighborhood of the node $v_j$. Since the MNEN method considers the position of a node and the scale of its neighbors as important factors in measuring the influence provided by the node, the $Ks$ value and degree value of the node are introduced when calculating $\Gamma(v_j)$ and $I(v_j)$. However, in actual networks, there is a numerical scale difference between the two. The degree value of a node usually has a larger order of magnitude. If the degree value is directly used for calculation, it will first dominate the calculation result of the influence provided by neighboring nodes, reducing the importance of the position reflected by the $Ks$ value; secondly, it will make the influence provided by neighboring nodes have an overly large proportion in the calculation of node influence, weakening the importance of the node's own influence. To avoid these problems, the logarithm of the node's degree is taken when calculating $\Gamma(v_j)$ and $I(v_j)$. The calculation formulas of $\Gamma(v_j)$ and $I(v_j)$ are as follows:

$$\Gamma(v_j) = K_S v_j^{1gd(v_j)/k_{max}} \qquad (4)$$

$$I(v_j) = \sum_{v_r \in EN(v_i, v_j)} K_s(v_r)^{1gd(v_r)/k_{max}} \qquad (5)$$

The formula for calculating $NI(v_j)$ is as follows:

$$NI(v_j) = \Gamma(v_j) + I(v_j) \qquad (6)$$

Where $k_{max}$ represents the maximum node degree, and $v_r$ is the node in the exclusive neighborhood of node $v_j$ relative to node $v_i$.

**Definition 4** (relevance between nodes): The final influence provided by the neighbor node $v_j$ of the node $v_i$ is related to the correlation $H(v_i, v_j)$ between the two nodes. This value is 0 because two nodes may exist without a common neighbor node; therefore, it requires improvement. The specific calculation is as follows:

$$H(v_i, v_j) = \frac{Nei(v_i) \cap Nei(v_j)| + 1}{nums} \qquad (7)$$

Where $Nei(v_i)$ is the set of neighbor nodes of node $v_i$, $Nei(v_j)$ is the set of neighbor nodes of node $v_j$, and nums represents the total number of nodes in the network.

**Definition 5** (actual influence provided by neighbor nodes): When calculating the influence that $v_i$'s neighbor node $v_j$ can actually provide it, if the initial influence value of node $v_j$ on $v_i$ is directly provided to $v_i$, the influence provided by neighboring nodes will be too great, and the node's importance will be diminished. When calculating the influence of a node in a practical situation, the node itself occupies a significant position, so it is necessary to reduce the influence provided by neighboring nodes. The MNEN method introduces the correlation between two nodes. Consequently, the initial value of node $v_j$'s influence is multiplied by the correlation between the two nodes. The specific calculation is as follows:

$$PNI\ (v_j) = NI\ (v_j) * H\ (v_i,\ v_j)$$
(8)

**Definition 6** (overall influence of a node): The overall influence of a node $v_i$ in the network is comprised of both its own influence and the influence that each of its neighbors can provide to it. The specific calculation is as follows:

$$K(v_i) = SI\ (v_i) + \sum_{v_j Nei(v_i)} PNI\ (v_j)$$
(9)

### Method process description

Based on the MNEN method, the influence value of a node is calculated. The entire procedure consists primarily of three components.

(i)  According to the degree of the node and the *Ks* value of the node, the influence of the node itself is calculated;

(ii) Calculate the influence that each of its neighbors can have on it. In this step, first, calculate the influence of a neighbor node, then determine the exclusive neighborhood of the neighbor node, and calculate the influence that the exclusive neighborhood can provide. Because this value is large, the correlation between nodes is introduced to reduce its proportion in the final influence calculation of nodes; the influence contribution of all neighbor nodes is added to obtain the final influence value.

(iii) Integrate the values obtained from the preceding two steps to assess the collective influence of the nodes, and subsequently arrange them accordingly.

The specific working process of this method is as follows:

```
MNEN algorithm:

Input:
  G=(V, E)
Calculate Ks value of all nodes
Calculate the maximum value k_max of all node degrees
For each node in set V v_i do
  Use formula (1) to calculate the influence SI(v_i)
  //Calculate neighbor node contribution I(v_j)
  For each node v_j in node v_i's neighbor node set do
    Use Formula (4) to calculate the influence of node v_j itself Γ(v_j)
    For each node v_r in the exclusive neighborhood of node v_j do
      The influence provided by computing node v_r
    endfor
    Use Formula (5) to calculate the influence I(v_j) provided by the exclusive neighborhood of node
    v_j
    Use Formula (6) to get the initial influence that neighbor node v_j can provide
    Use Formula (7) to get the correlation between v_i and v_j
    Use Formula (8) to obtain PNI(v_j)
```

```
    endfor
  Use Formula (9) to obtain K(vᵢ)
endfor
//All nodes in the network are sorted by influence value
Return Rank(V)
Output:
  Influence value sorting of all nodes
```

## Model network analysis

As can be seen from Formula (9), in the MNEN method, the influence of a node is determined by its own influence and the influence provided by all of its neighbors. When a node is located at the core of the network, has many neighbors, and its neighboring nodes are relatively important, the influence of the node is greater. In <u>Fig 3</u>, $v_4$ is an example.

First, calculate $SI(v_4)$ of the node $v_4$ itself. Since $Ks(v_4)=3$ and $d(v_4)=6$, then $SI(v_4)=Ks(v_4)+ d(v_4)=9$.

Second, calculate the influence that the neighbor nodes of node $v_4$ have on it. According to formula (6), the influence provided by the neighbor nodes consists of their own influence and the influence of their neighbor nodes. The neighbor node set of $v_4$ is $\{v_1, v_2, v_3, v_5, v_{11}, v_{12}\}$. Take node $v_3$ as an example to illustrate the calculation process of the influence provided by neighboring nodes. The neighbor node set of $v_3$ is $\{v_1, v_2, v_4, v_9, v_{10}\}$. Then, the common neighbor node set $CN(v_4, v_3)=\{v_1, v_2\}$ of $v_3$ and $v_4$. Nodes $v_1$ and $v_2$ are both neighbors of $v_4$ and $v_3$, which leads to the situation of repeated calculation of node influence. Therefore, when calculating the influence provided by node $v_3$, the common neighbor nodes $v_1$ and $v_2$ with $v_4$ are removed, and only the unique neighbor nodes of $v_3$ are retained, that is, the exclusive neighborhood

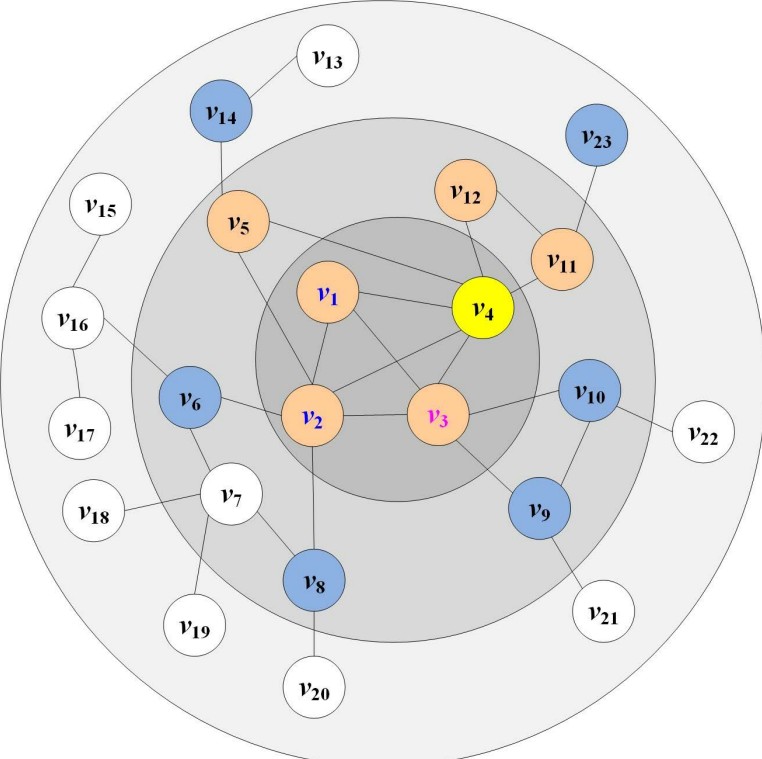

**Fig 3. Schematic diagram of network structure.**

$EN(v_4, v_3)$ of $v_3$ relative to $v_4$ is $\{v_4, v_9, v_{10}\}$. This method can avoid the problem of inaccurate results caused by the repeated calculation of the influence of nodes that are connected to many other nodes. In Fig 3, node $v_4$ is shown in yellow, one-hop neighbor nodes are shown in orange, and the exclusive neighborhood of one-hop neighbor nodes relative to $v_4$ (excluding $v_4$) is shown in blue.

The $Ks$ value and degree value of node $v_3$ are $Ks(v_3)=3$ and $d(v_3)=5$. In Fig 3, the maximum value $k_{max}$ of all node degrees is 6, then the self-influence of node $v_3$: $\Gamma(v_3) = K_s(v_3)^{1gd(v_3)/k_{max}} = 3^{1g(5)/6} = 1.137$, the calculation formula of influence provided by the exclusive neighborhood of node $v_3$ is $I(v_3)=2.113$. The influence value that $v_3$ can provide is $NI(v_3)=\Gamma(v_3)+I(v_3)=1.137+2.113=3.25$. Since the correlation between $v_3$ and $v_4$ is $(2+1)/23=0.1304$, the influence value that $v_3$ can actually provide is $0.13*3.25=0.424$. By analogy, calculate the actual influence value that all neighbor nodes of node $v_4$ can provide: $PNI(v_1)=0.142$, $PNI(v_2)=0.568$, $PNI(v_5)=0.179$, $PNI(v_{11})=0.179$, and $PNI(v_{12})=0.09$, the influence value of node $v_4$ is 10.582. The influence values of the core area nodes $v_1$, $v_2$, and $v_3$ in the network were calculated using this method, which are 7.549, 10.581, and 9.625 respectively. More detailed influence values of the nodes can be obtained through a Python program.

Python is used to implement the SIR, BC, CC, EC, K-shell, PR, Gravity, PL, ELKSS, GIN, KBKNR, GSM, ECRM, and MNEN algorithms. Then, obtain the sequences of the first 10 nodes in the model network. Set the SIR model contagion probability $\alpha=0.01$, and the number of iteration runs $t=1000$. The sorting results of each algorithm and SIR are displayed in Table 1.

According to Table 1 in the example network, the MNEN algorithm identified nodes $v_4$, $v_2$, $v_1$, and $v_3$ as having relatively high influence. This table presents the top 10 nodes identified by the SIR model and various methods. Nodes in the SIR column are marked in bold; if the top 10 node sequence identified by a method is exactly the same as that of SIR, the corresponding nodes are also marked in bold; if the top 10 nodes identified by a method exist in SIR's top 10 but the sequence is not the same, the nodes are marked in italics; nodes that do not enter the SIR top 10 list are not specially marked.

Due to the relatively small scale of the example network, all algorithms achieved relatively satisfactory results. Among the top 10 node sequences of each algorithm and SIR, the first 6 nodes of the MNEN method were consistent with the SIR sequence, with only 1 node not appearing in the top 10 nodes of the SIR sequence. In the EC method, the first 4 nodes were consistent with the SIR sequence, but 2 nodes were different from SIR. Although only the first 3 nodes of the PL method were consistent with SIR, only 1 node was different from SIR.

## Time complexity analysis

The time complexity consists primarily of three components. In the first step, the K-shell algorithm is used to calculate the $Ks$ value of each node, with a time complexity of $O(n)$; the degree value of each node is then calculated, with a time

**Table 1. Algorithm sorting results (the last column is sorted by SIR).**

| BC | CC | EC | K-shell | PR | Gravity | PL | ELKSS | GIN | KBKNR | GSM | ECRM | MNEN | SIR |
|----|----|----|---------|----|---------|-----|-------|-----|-------|-----|------|------|-----|
| 2 | 2 | 4 | 1 | 2 | 2 | 4 | 2 | 2 | 2 | 2 | 2 | 4 | 4 |
| 6 | 4 | 2 | 2 | 4 | 4 | 2 | 4 | 4 | 4 | 4 | 4 | 2 | 2 |
| 3 | 3 | 3 | 3 | 7 | 3 | 3 | 3 | 3 | 3 | 3 | 3 | 3 | 3 |
| 4 | 6 | 1 | 4 | 3 | 1 | 7 | 1 | 1 | 1 | 1 | 1 | 1 | 1 |
| 7 | 1 | 5 | 5 | 16 | 5 | 8 | 5 | 6 | 7 | 5 | 5 | 7 | 7 |
| 8 | 5 | 11 | 6 | 8 | 6 | 11 | 6 | 5 | 6 | 6 | 6 | 5 | 5 |
| 16 | 8 | 12 | 8 | 6 | 8 | 6 | 11 | 8 | 5 | 8 | 8 | 11 | 6 |
| 5 | 7 | 9 | 9 | 9 | 11 | 5 | 8 | 11 | 8 | 11 | 11 | 9 | 11 |
| 9 | 11 | 10 | 10 | 10 | 9 | 9 | 9 | 9 | 9 | 9 | 9 | 10 | 8 |
| 10 | 9 | 6 | 11 | 11 | 10 | 10 | 10 | 10 | 10 | 10 | 10 | 6 | 10 |

complexity of $O(n)$. The second step is to calculate the influence of the node itself by adding the node's degree value to the $Ks$ value, with an $O(n)$ time complexity. The third step is to identify the one-hop neighbor node and the two-hop neighbor node, calculate the influence that the neighbor node can provide, and the time complexity is $O(<m><k>)$. To sum up, the time complexity of this method is $O(n<k>^2)$.

## Experimental evaluations

In this section, we perform simulation experiments with the MNEN method and 12 key node identification methods over 10 networks by SIR modeling, and then the obtained simulation results are compared and analyzed. The 12 influential node identification methods are BC, CC, EC, K-shell, PR, Gravity, PL, ELKSS, GIN, KBKNR, GSM, and ECRM.

### Data description

In the experiment of the influential node identification performance of the MNEN method, ten real networks are used for validation: the Protein network, the Email network, the Ca-Erdos network, the Ca-Astroph network, the Karate network, the Dolphins network, the USAir network, the Euroroad network, the Ca-GrQc network, and the Powergrid network. These datasets are available at http://konect.cc/networks/ and http://networkrepository.com/networks.php.

Table 2 shows the statistics of relevant characteristics of the above ten datasets.

In Table 2, $|V|$ represents the number of nodes, $|E|$ is the number of edges, $<k>$ is the average degree, $<cc>$ is the average clustering coefficient, and $k_{max}$ represents the maximum node degree in the network.

### Evaluation indicators

In this paper, the SIR propagation model [35] is employed to evaluate the node sorting sequences generated by various algorithms operating on various networks. Even though there are many evaluation methods for influential node identification effects at present, the SIR model is still widely used.

There are three types of nodes in this model: susceptible node S, infectious nodes I, and recovered node R. The susceptible node is not infected but is easily infected when connected to the infected node; the infected node has been infected but has not recovered and can infect other nodes, and the recovery node has been infected and has recovered. It is no longer infected. When the network is operational, the infected node has a probability Infect the uninfected node connected to it, turning it into an infected node, and the infected node becomes a recovery node with a certain probability β. Moreover, this procedure is repeated repeatedly. Various nodes can be used as seed nodes to simulate the infection process until obtaining the infection influence of all nodes.

**Table 2. Feature statistics of the ten real networks.**

| Datasets | $|V|$ | $|E|$ | $<k>$ | $<cc>$ | $k_{max}$ |
|---|---|---|---|---|---|
| Protein | 1870 | 2277 | 2.435 | 0.171 | 56 |
| Email | 1133 | 5451 | 9.622 | 0.254 | 71 |
| Ca-Erdos | 5094 | 7515 | 2.951 | 0.279 | 61 |
| Ca-Astroph | 18771 | 198050 | 21.102 | 0.677 | 236 |
| Karate | 34 | 78 | 4.588 | 0.588 | 17 |
| Dolphins | 62 | 159 | 5.129 | 0.303 | 12 |
| USAir | 1226 | 2410 | 3.931 | 0.073 | 34 |
| Euroroad | 1174 | 1417 | 2.414 | 0.02 | 5 |
| Ca-GrQc | 4158 | 13422 | 6.456 | 0.665 | 81 |
| Powergrid | 4941 | 6594 | 2.669 | 0.107 | 19 |

Twelve algorithms and the SIR model are used to calculate the influence of each node. In addition, the nodes are sorted according to the influence from high to low. The Kendall τ coefficient [36] is then used to measure the sequences obtained by each algorithm and the SIR model. Suppose the network contains m nodes. The node sequence obtained by an algorithm is represented by $R=(r_1, r_2,..., r_m)$, and the node sequence obtained by the SIR model is represented by $S=(s_1, s_2,..., s_m)$. If the sequence pair $(r_i, s_i)$ $(i=1, 2,..., m)$ in R and S satisfies both $r_i>r_j$ and $s_i>s_j$, or $r_i<r_j$ and $s_i<s_j$, then this sequence pair is consistent. If $r_i>r_j$ and $s_i<s_j$, or $r_i<r_j$ and $s_i>s_j$, the sequence pair is inconsistent. If $r_i=r_j$ and $s_i=s_j$, the sequence pair is neither consistent nor inconsistent.

$$\tau(R,\ S) = \frac{2(m_c - m_d)}{m(m-1)}$$

(10)

In Formula (10), $m_c$ represents the number of consistent sequence pairs in two sequences, and $m_d$ is the number of inconsistent sequence pairs in two sequences. The $\tau(R,\ S)$ value is proportional to the accuracy of the acquired sequence.

The Jaccard similarity coefficient is adopted to quantitatively analyze the consistency degree of each algorithm with the SIR model in the identification of key nodes. For a given number of key nodes m, the first m nodes identified by each algorithm form set A, and the first m nodes identified by the SIR model form the reference set B. The calculation formula of the Jaccard similarity coefficient is as follows:

$$J(A, B) = \frac{|A \cap B|}{|A \cup B|}$$

(11)

The similarity between the recognition results of each algorithm and the SIR model was objectively measured by calculating the ratio of the intersection to the union of the two sets. The Jaccard coefficient ranges from [0, 1], with a value closer to 1 indicating a higher consistency in the key node sets identified by the two methods, and a value closer to 0 indicating a greater difference.

### Experimental performance analysis

Using a Python program to implement MNEN, 12 comparison methods, and SIR models, the influence of nodes is calculated. According to the SIR model, the infection effect of a node is determined by the number of infected and recovery nodes. Due to the randomness of the model, multiple SIR iterations are used to calculate the average value in order to obtain relatively stable and reliable data. Due to the large number of nodes, edges, and long running time of the Ca-Astroph network, the number of iterations is set to 100, while the number of iterations for the remaining nine networks is set to 1000. At the same time, since GSM and Gravity have significantly longer running times than other methods in the Ca-Astroph network, the verification process data for the subsequent experiments will not be presented.

**Jaccard similarity coefficient.** First, the influence of nodes in each network is calculated using each method and SIR model, and the node sequence is obtained in descending order. The infection probability α in the SIR model is set to 0.05. Then, the Jaccard similarity coefficient of each algorithm and the SIR model on the top 15 nodes is calculated using formula (11), and finally presented in the form of a heat map, as shown in Fig 4.

Fig 4 illustrates the level of consistency between each algorithm and the SIR model in identifying key nodes across different networks.

As observed in Fig 4, in the Protein and Powergrid networks, the Jaccard similarity between the top 15 nodes identified by the MNEN method and the SIR model is relatively high, suggesting that the MNEN method effectively identifies key nodes with greater influence within the network, achieving an ideal identification performance. In the Karate network, due to its smaller scale, all algorithms exhibit relatively good identification performance. In the Ca-Erdos, Dolphins, USAir, Euroroad, and Ca-GrQc networks, most algorithms demonstrate satisfactory identification performance. In the Email network, the MNEN method performs slightly less effectively compared to the EC, ELKSS, and ECRM methods. In the

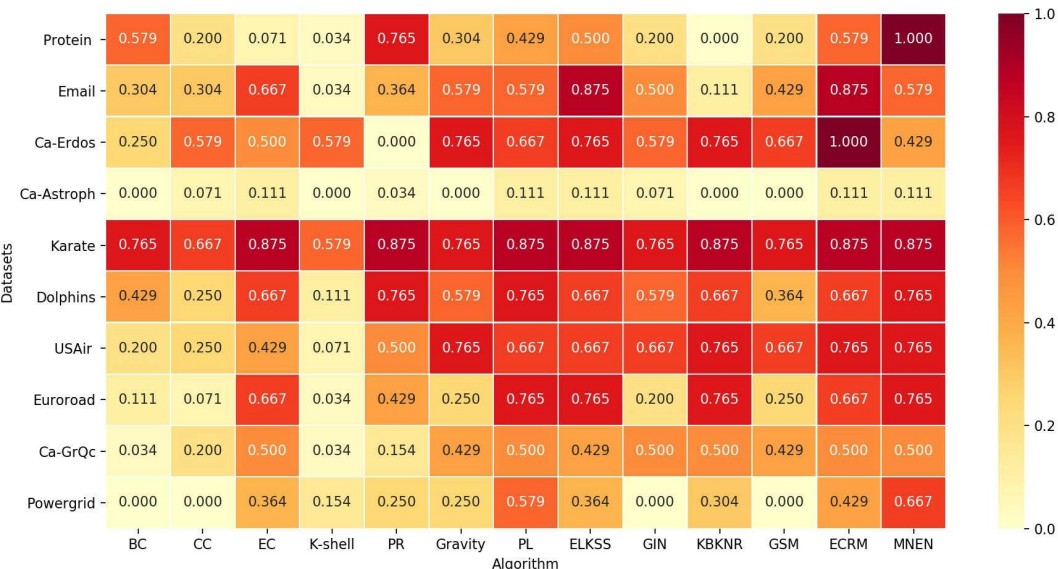

**Fig 4. Heatmap of Jaccard similarity between each algorithm and the top 15 nodes of SIR.**

Ca-Astroph network, owing to its large scale, the identification performance of all methods is less than optimal. Since this similarity reflects whether the nodes within a certain range are the same, it cannot show whether the node sequences are consistent. Therefore, a table format was adopted to list the node sequences intuitively.

Due to space limitations, here we take the Dolphins, USAir, and Powergrid networks as examples. In Tables 3–5, the top 10 nodes obtained by all methods and SIR in these three networks are displayed. For consistency, Tables 3–5 all adopt the same format rules as Table 1.

According to Table 3, in the Dolphins network, the recognition effects of various algorithms are different. The EC, ELKSS, KBKNR, ECRM, and MNEN methods perform relatively well on this network, with the top 4 nodes identified being consistent with the sequence of the SIR model. PR and PL also achieved acceptable results, with multiple nodes among the top 10 being consistent with the results of the SIR model. However, the BC, CC, and K-shell methods did not perform well, especially the K-shell method. Due to its coarse-grained nature, it could only identify 2 nodes among the top 10 as being consistent with the SIR model.

**Table 3. The top 10 node sequences of different algorithms in the Dolphins network.**

| BC | CC | EC | K-shell | PR | Gravity | PL | ELKSS | GIN | KBKNR | GSM | ECRM | MNEN | SIR |
|----|----|----|---------|----|---------|----|-------|-----|-------|-----|------|------|-----|
| 37 | 37 | **15** | 1 | **15** | **15** | **15** | **15** | *38* | **15** | **15** | **15** | **15** | **15** |
| 2 | 41 | **38** | 11 | **18** | **38** | *46* | **38** | *15* | **38** | **38** | **38** | **38** | **38** |
| 41 | *38* | **46** | *15* | 52 | **46** | 38 | **46** | *46* | **46** | 41 | **46** | **46** | **46** |
| *38* | *21* | **34** | 16 | 58 | *21* | 34 | **34** | 41 | **34** | *21* | **34** | **34** | **34** |
| 8 | *15* | 51 | 41 | 38 | 41 | 52 | 41 | *34* | 52 | 37 | 41 | 52 | **21** |
| *18* | 2 | *30* | 43 | *46* | *34* | 58 | 51 | *21* | *21* | *46* | 51 | *30* | **52** |
| *21* | 29 | *52* | 48 | *34* | 37 | *21* | *30* | 37 | *30* | *34* | *30* | *18* | **58** |
| 55 | 8 | 17 | 2 | *30* | 51 | 14 | *52* | 51 | 41 | 51 | *52* | *21* | **30** |
| *52* | *34* | 41 | *18* | 14 | *30* | *30* | *21* | 19 | *18* | 2 | *21* | 58 | **14** |
| *58* | 9 | 22 | 20 | 2 | 2 | *18* | 17 | *30* | *58* | *30* | 17 | 41 | **18** |

**Table 4. The top 10 node sequences of different algorithms in USAir network.**

| BC | CC | EC | K-shell | PR | Gravity | PL | ELKSS | GIN | KBKNR | GSM | ECRM | MNEN | SIR |
|---|---|---|---|---|---|---|---|---|---|---|---|---|---|
| **68** | **68** | *113* | 1 | *312* | *52* | *52* | *52* | **68** | **68** | **68** | *52* | *312* | **68** |
| **52** | **52** | *52* | 2 | *68* | *68* | *113* | *68* | **52** | **52** | **52** | *68* | *52* | **52** |
| 213 | 148 | *47* | 3 | *52* | 148 | *68* | *47* | 148 | *113* | 148 | *113* | *47* | **312** |
| *312* | 135 | *34* | 4 | 212 | 135 | *47* | *113* | 135 | *89* | 135 | *47* | *113* | **113** |
| 135 | 711 | *68* | 6 | *113* | *44* | *44* | *44* | *44* | 187 | *44* | *44* | *89* | **89** |
| 136 | 424 | 116 | 10 | *187* | *113* | *34* | *34* | 116 | *47* | 116 | *34* | *187* | **47** |
| 212 | 136 | *109* | 14 | 523 | 110 | *109* | 116 | 124 | *44* | 124 | 116 | *47* | **44** |
| 660 | 617 | 102 | 16 | *89* | *89* | 116 | 46 | 110 | 135 | 110 | *109* | *44* | **187** |
| 523 | 20 | 82 | 18 | 604 | 116 | 102 | 102 | 46 | *109* | 46 | 102 | *109* | **109** |
| 221 | 116 | 66 | 20 | 218 | 537 | 82 | *89* | *113* | 110 | *113* | 46 | 135 | **34** |

**Table 5. The top 10 node sequences of different algorithms in Powergrid network.**

| BC | CC | EC | K-shell | PR | Gravity | PL | ELKSS | GIN | KBKNR | GSM | ECRM | MNEN | SIR |
|---|---|---|---|---|---|---|---|---|---|---|---|---|---|
| 651 | 1378 | *4422* | *4422* | 602 | *4422* | 4436 | *4422* | 2781 | *4422* | 2781 | *4436* | **2847** | **2847** |
| 559 | 1678 | *4436* | 4415 | 932 | *4452* | 4422 | 4436 | 2685 | *4417* | 2685 | *4422* | **602** | 602 |
| 1365 | 2944 | *4419* | *4452* | 3411 | *4417* | 4453 | *4452* | 559 | *4452* | 559 | *4419* | **4436** | **4436** |
| 2824 | 1377 | *4417* | 4453 | *2847* | 4453 | *2847* | 4453 | 2944 | 4453 | 2944 | *4434* | *4422* | **558** |
| 2685 | 2781 | *4452* | 4427 | 1210 | 4427 | *4452* | 4419 | 2824 | 4427 | 1378 | 4438 | 4452 | **4417** |
| 1324 | 1365 | 4453 | 4428 | 691 | 4421 | *4434* | 4417 | 2956 | 4415 | 1324 | *4417* | 4453 | **4452** |
| 1378 | 1368 | 4427 | 4451 | 2287 | 4425 | *4417* | 4434 | 651 | 4421 | 651 | *4452* | *4417* | **4434** |
| 1213 | 1380 | 4421 | 4454 | 2865 | 4428 | *2926* | 4427 | 1378 | 4428 | 1365 | 4453 | *2926* | **4422** |
| 433 | 2685 | *4434* | *4417* | 2554 | 4415 | 2543 | 4438 | 2782 | 4425 | 1678 | 4437 | *558* | **2926** |
| 2781 | 2795 | 4438 | 4418 | 3930 | *4436* | 4438 | 4421 | 1678 | 4454 | 2824 | *2847* | 932 | **4419** |

As shown in Table 4, in the USAir network, the recognition effects of various algorithms are different.

According to Table 4, it can be seen that the first two node sequences of KBKNR, BC, CC, GIN, GSM and SIR are consistent. Among them, the recognition effect of KBKNR is relatively ideal, with 8 out of the top 10 nodes being the same as those of SIR. Although the node sequences of MNEN, PL and ECRM methods are not consistent with that of SIR, many of the top 10 nodes they identified are the same as those of SIR, thus they also have certain recognition effects. However, PR and Gravity methods have no consistent nodes with SIR, and the number of nodes that are not the same as the top 10 nodes of SIR is relatively large, so their recognition effects are not very ideal.

As shown in Table 5, in the Powergrid network, the identification effects of various algorithms are different. The first three nodes identified by MNEN are consistent with the SIR model sequence, and among the top 10 nodes, 8 are the same as those in the SIR model. The PL, ECRM, EC, and ELKSS methods do not have any nodes that are consistent with the SIR model sequence, but among the top 10 nodes, more than 6 are the same as those in the SIR model. The identification effects of K-shell, PR, BC, and CC are not very satisfactory, especially for BC and CC. This might be due to the large scale of the network, as none of the top 10 nodes are the same as those in the SIR model.

**Kendall values under different infection probabilities.** If the node sequence obtained by a particular method in a network is $R$, then the node sequence obtained by the SIR model is $S$. In addition, the infection probability of the SIR model is set between 0.01 and 0.1, with an increment of 0.01, a total of 10 values are obtained, and a sequence can be obtained for each probability; all sequences can be represented by $R=\{R_1, R_2,..., R_m\}$ (m = 10),

and the sequence $S$ and $R_i$ ($1 \leq i \leq 10$) are calculated using Formula (10) to obtain Kendall τ values under different infection probabilities. Taking the Protein network and the Ca-Erdos network as examples, the Kendall τ values of different algorithms are shown.

In Table 6, due to the existence of isolated nodes in the Protein network, the KBKNR method cannot be applied to this network. Among the remaining 12 methods, the MNEN method has the best effect, and the value exhibits an upward trend; thus, the node sequence obtained by this method is more consistent with that obtained by SIR.

According to Table 7, it can be seen that different infection probabilities have an impact on the Kendall τ ranking consistency of each method.

When the infection probability is 0.01, the KBKNR method performs the best; when the infection probability is within the range of [0.02, 0.05], the MNEN method shows better ranking consistency; when the infection probability is 0.06, both the ELKSS and ECRM methods start to show advantages; when the infection probability is within the range of [0.07, 0.1], the performance of the Gravity, ELKSS, GIN, GSM and ECRM methods is superior to that of the MNEN method.

The Kendall τ values of 13 methods in different networks are shown graphically, as shown in Fig 5.

Since KBKNR method cannot be applied to networks with isolated nodes, there is no polyline corresponding to this method in the Protein network in Fig 5, and MNEN method has the best effect; in the other nine networks, MNEN is compared with KBKNR method, and the values of Kendall τ are relatively close.

**Table 6. Kendall τ values of different algorithms in Protein network.**

| α | BC | CC | EC | K-shell | PR | Gravity | PL | ELKSS | GIN | KBKNR | GSM | ECRM | MNEN |
|---|----|----|----|---------|----|---------|----|-------|-----|-------|-----|------|------|
| **0.01** | 0.521 | 0.369 | 0.330 | 0.444 | 0.452 | 0.477 | 0.523 | 0.442 | 0.371 | – | 0.378 | 0.443 | 0.664 |
| **0.02** | 0.528 | 0.426 | 0.388 | 0.450 | 0.424 | 0.535 | 0.497 | 0.510 | 0.430 | – | 0.437 | 0.512 | 0.729 |
| **0.03** | 0.531 | 0.478 | 0.437 | 0.451 | 0.379 | 0.583 | 0.452 | 0.566 | 0.482 | – | 0.488 | 0.567 | 0.772 |
| **0.04** | 0.524 | 0.521 | 0.482 | 0.449 | 0.336 | 0.620 | 0.413 | 0.615 | 0.527 | – | 0.533 | 0.617 | 0.794 |
| **0.05** | 0.514 | 0.555 | 0.518 | 0.445 | 0.297 | 0.649 | 0.378 | 0.655 | 0.562 | – | 0.567 | 0.657 | 0.812 |
| **0.06** | 0.505 | 0.585 | 0.547 | 0.443 | 0.265 | 0.673 | 0.350 | 0.690 | 0.594 | – | 0.598 | 0.691 | 0.820 |
| **0.07** | 0.497 | 0.607 | 0.572 | 0.440 | 0.243 | 0.689 | 0.337 | 0.714 | 0.616 | – | 0.620 | 0.715 | 0.816 |
| **0.08** | 0.486 | 0.627 | 0.590 | 0.434 | 0.219 | 0.704 | 0.316 | 0.739 | 0.637 | – | 0.642 | 0.740 | 0.810 |
| **0.09** | 0.477 | 0.647 | 0.610 | 0.429 | 0.198 | 0.717 | 0.296 | 0.761 | 0.657 | – | 0.661 | 0.762 | 0.808 |
| **0.1** | 0.469 | 0.662 | 0.628 | 0.427 | 0.181 | 0.727 | 0.282 | 0.780 | 0.674 | – | 0.676 | 0.779 | 0.800 |

**Table 7. Kendall τ values of different algorithms in Ca-Erdos network.**

| α | BC | CC | EC | K-shell | PR | Gravity | PL | ELKSS | GIN | KBKNR | GSM | ECRM | MNEN |
|---|----|----|----|---------|----|---------|----|-------|-----|-------|-----|------|------|
| **0.01** | 0.366 | 0.369 | 0.310 | 0.426 | 0.422 | 0.477 | 0.360 | 0.415 | 0.372 | 0.553 | 0.372 | 0.438 | 0.551 |
| **0.02** | 0.369 | 0.470 | 0.396 | 0.436 | 0.361 | 0.580 | 0.251 | 0.530 | 0.474 | 0.678 | 0.473 | 0.556 | 0.678 |
| **0.03** | 0.364 | 0.521 | 0.447 | 0.436 | 0.318 | 0.631 | 0.177 | 0.600 | 0.528 | 0.744 | 0.527 | 0.630 | 0.745 |
| **0.04** | 0.357 | 0.576 | 0.501 | 0.431 | 0.271 | 0.676 | 0.140 | 0.663 | 0.585 | 0.763 | 0.583 | 0.691 | 0.765 |
| **0.05** | 0.346 | 0.620 | 0.548 | 0.422 | 0.223 | 0.704 | 0.107 | 0.717 | 0.632 | 0.764 | 0.630 | 0.741 | 0.768 |
| **0.06** | 0.334 | 0.660 | 0.597 | 0.414 | 0.176 | 0.730 | 0.115 | 0.766 | 0.675 | 0.729 | 0.673 | 0.782 | 0.735 |
| **0.07** | 0.320 | 0.693 | 0.649 | 0.402 | 0.131 | 0.747 | 0.115 | 0.800 | 0.713 | 0.692 | 0.712 | 0.796 | 0.699 |
| **0.08** | 0.302 | 0.726 | 0.705 | 0.385 | 0.076 | 0.758 | 0.122 | 0.833 | 0.752 | 0.639 | 0.750 | 0.804 | 0.646 |
| **0.09** | 0.287 | 0.742 | 0.750 | 0.371 | 0.040 | 0.758 | 0.127 | 0.841 | 0.772 | 0.596 | 0.772 | 0.789 | 0.604 |
| **0.1** | 0.274 | 0.757 | 0.787 | 0.357 | 0.006 | 0.757 | 0.131 | 0.836 | 0.791 | 0.557 | 0.792 | 0.770 | 0.565 |

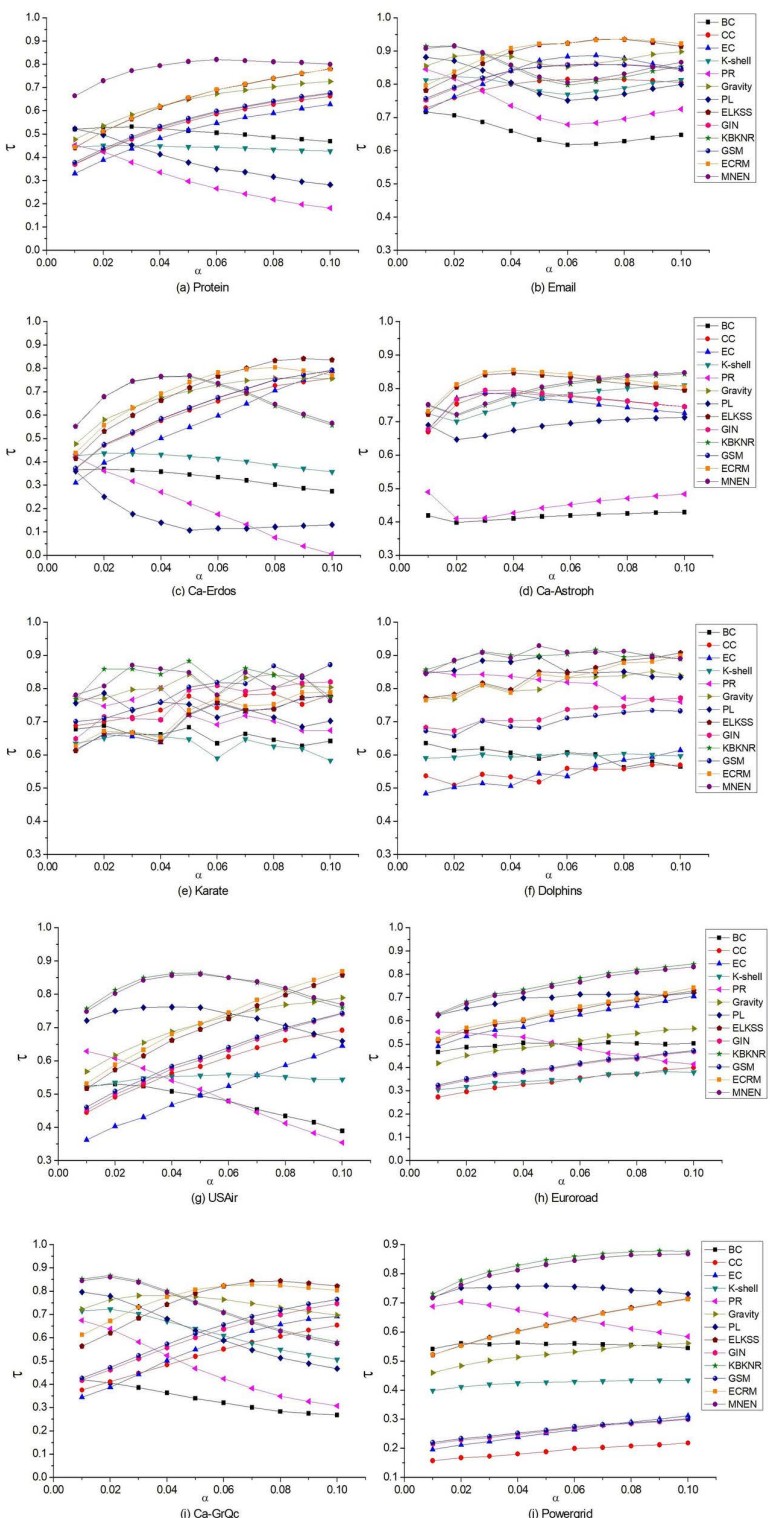

**Fig 5. The Kendall τ values obtained by comparing the sequence results of the 13 methods and the SIR model on ten networks, and the infection probability is [0.01, 0.1].**

In Protein, Euroroad, and Powergrid networks, the MNEN method has higher Kendall τ values from 0.01 to 0.1 than other algorithms; in the Dolphins network, it is only lower than ELKSS at 0.1; in the USAir network, the MNEN method is higher than other algorithms from 0.01 to 0.08, lower than ECRM and ELKSS at 0.09, lower than ECRM, ELKSS, and Gravity at 0.1; in the Karate, Email, Ca-GrQc, Ca-Erdos, and Ca-Astroph networks, the recognition effect of the MNEN method is generally average.

The Kendall τ value indicates the consistency between the node sequences obtained by this method and by the SIR model. Therefore, in ten different networks, the performance of MNEN is relatively stable and has achieved a good identification effect.

**The infection capability of the algorithm on the SIR model.** The experimental results are represented as a stacked graph, this is shown in Fig 6.

The sequences obtained by various algorithms are compared to those obtained by SIR in order to determine the infectious ability sequence of nodes in the SIR model.

Since there are many nodes in the Ca-Erdos and Ca-Astroph networks, the infection probability of the nodes is set to 0.01, while the infection probability of the nodes in the other eight networks is 0.1. In this experiment, each node is considered a seed node that infects its neighbors with a predetermined probability.

The Karate and Dolphins networks are small, so the infectivity data for each node is displayed directly in Fig 6. In the remaining eight networks, the infectivity data is presented as log10 due to the networks' size. When the curve shows a smooth downward trend from left to right, the higher the consistency between the node importance sequence obtained by each algorithm and the sequence obtained by the SIR model, the less the area of the curve fluctuates. In the Email, Ca-Erdos, Karate, Dolphins, Euroroad, and Powergrid networks, MNEN has the greatest effect; in the Protein network, MNEN is inferior to the ECRM method and is close to that of the ELKSS method; in the Ca-Astroph and USAir networks, the performance of the MNEN method is average compared with the ECRM, ELKSS, and PL methods; in the Ca-GrQc network, the overall curves of all algorithms vary substantially, but the curves of the GSM, GIN, and EC methods have relatively fewer waves, and their performance is slightly better than that of the MNEN method.

**The infective capacity of the top 10 nodes.** In this experiment, the first ten nodes of various methods are used as seed nodes, the time step t is set between 1 and 30, the number of infected and recovered nodes after infection of ten nodes at time t is calculated, and the infectivity of nodes is determined based on this number. In order to achieve a significant infection effect, given the small size of the Karate and Dolphins networks, the infection probability is set to 0.5. Since Ca-Astroph is small, if the probability of infection is too high, it will rapidly infect all network nodes. It is difficult to differentiate the infection effect. Therefore, the infection probability is set to 0.01, and the infection probability for the remaining seven networks is set to 0.1. The recovery probability for all networks is set to 1.

Select the top ten nodes in ten networks using 13 methods, and use them as seed nodes in the SIR model to calculate infection ability. Fig 7 depicts the experimental data.

Fig 7 demonstrates that in ten networks, the SIR Model is used to calculate the infection rate of the top ten nodes chosen by all methods over time. With the change of time step t, the infectivity tends to become increasingly stable. In different networks, the t value at which infectivity tends to stabilize differs, but it reaches a stable state before t = 10 in most cases.

According to Fig 7 in the Protein, Email, Ca-Erdos, Ca-Astroph, Dolphins, USAir, Euroroad, and Powergrid networks, the MNEN method's first ten nodes have a higher infection rate than other methods. In the Karate network, the BC method has the best overall effect, followed by the MNEN method, while the effects of the other methods are relatively close; in the Ca-GrQc network, MNEN method has a good effect before time step t = 2, but the infection effect is lower than BC, CC, and PR beginning at time step t = 3, which is close to other methods. Consequently, the MNEN has the greatest overall impact of the ten networks.

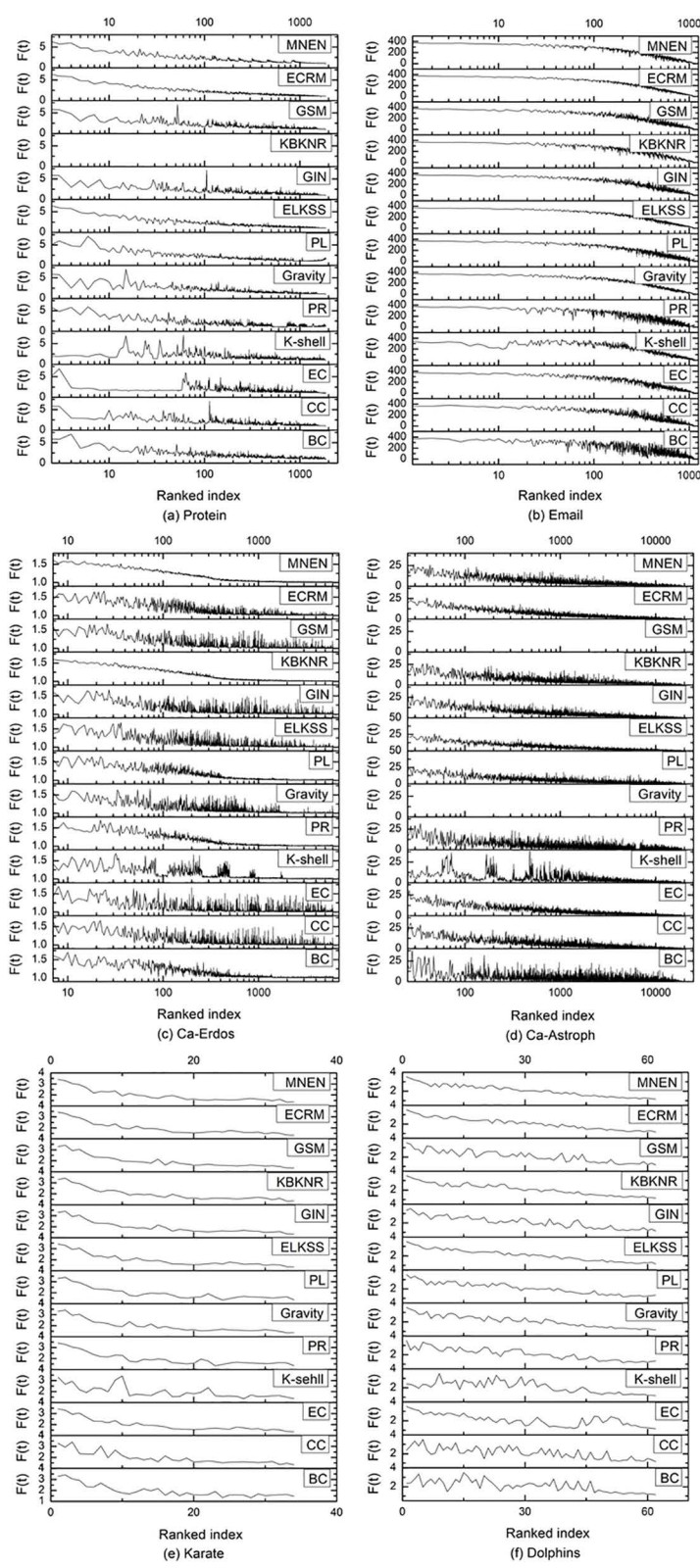

(a) Protein

(b) Email

(c) Ca-Erdos

(d) Ca-Astroph

(e) Karate

(f) Dolphins

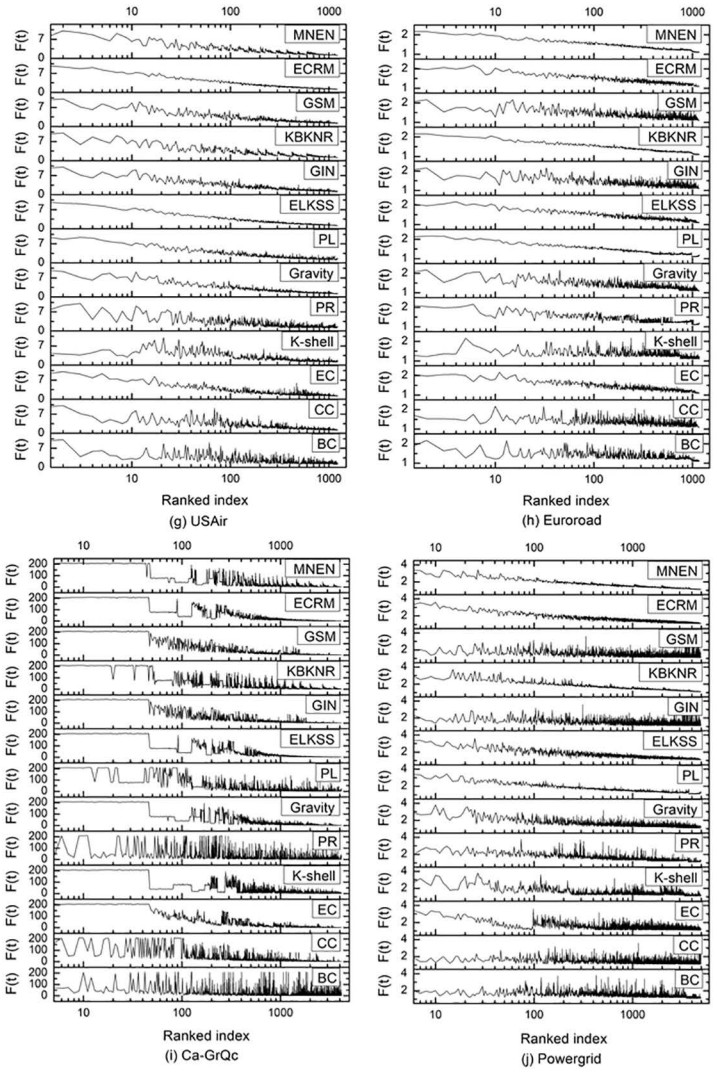

**Fig 6. The infection ability of a single node in different algorithms, Ranked index is the node sequence index, F(t) is the number of nodes infected by a node at time t, and the number of recovered nodes infected by it.**

## Conclusion

This paper proposes a method for identifying influential nodes in complex networks based on multi-order neighbors and exclusive neighbors, based on research on existing key node methods. This method comprehensively considers the contributions of the node and its neighbors. When calculating a node's own influence, the node's own degree and $Ks$ value are introduced. The contribution of neighbor nodes is primarily comprised of two aspects: calculating the contribution of neighbor nodes based on their degree and $Ks$. Calculate the influence provided by the node with two hops under certain conditions based on the exclusive neighborhood of the neighbor node, and then rank the nodes. To verify the influential node identification effect of the MNEN, a widely used SIR model is presented, experiments are conducted with 12 methods in ten real networks, and experimental data are analyzed. Experiments demonstrate that the method performs well on most networks, and that the first ten identified key nodes have a significant impact; however, the effect on individual

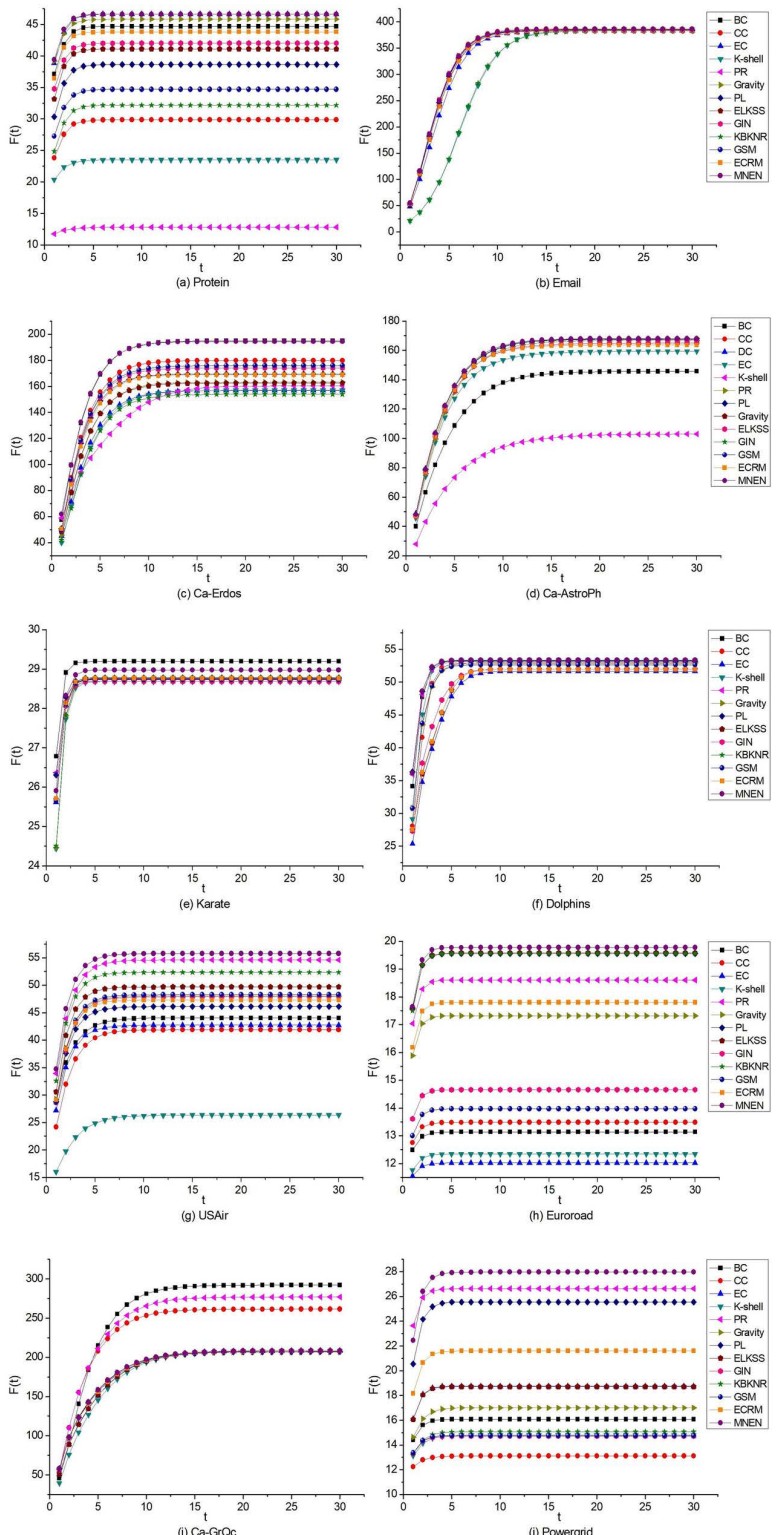

**Fig 7. The infectivity of the first ten nodes identified by 13 algorithms in 10 networks in the SIR model, F(t) is the infectivity at time t.**

networks is less than that of other algorithms. In future research, this method and the network with poor performance will be investigated in depth to determine the underlying causes and enhance the current methods.

## Author contributions

**Data curation:** Zejun Sun.

**Investigation:** Bohan Sun.

**Methodology:** Feifei Wang.

**Validation:** Guan Wang.

**Visualization:** Zejun Sun.

**Writing – original draft:** Feifei Wang.

**Writing – review & editing:** Feifei Wang.

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
