## [Decision Letter · Decision Letter 0]

2 Jun 2025

Dear Dr. Sun,

Thank you for submitting your manuscript to PLOS ONE. After careful consideration, we feel that it has merit but does not fully meet PLOS ONE’s publication criteria as it currently stands. Therefore, we invite you to submit a revised version of the manuscript that addresses the points raised during the review process.

We look forward to receiving your revised manuscript.

Kind regards,

Giridhar Maji, Ph.D.

Academic Editor

PLOS ONE

Journal Requirements:

3. Please note that your Data Availability Statement is currently missing the repository name and/or the DOI/accession number of each dataset OR a direct link to access each database. If your manuscript is accepted for publication, you will be asked to provide these details on a very short timeline. We therefore suggest that you provide this information now, though we will not hold up the peer review process if you are unable.

Reviewers' comments:

Reviewer's Responses to Questions

**Comments to the Author**

1. Is the manuscript technically sound, and do the data support the conclusions?

Reviewer #1: Yes

Reviewer #2: Partly

2. Has the statistical analysis been performed appropriately and rigorously?

Reviewer #1: Yes

Reviewer #2: Yes

3. Have the authors made all data underlying the findings in their manuscript fully available?

Reviewer #1: Yes

Reviewer #2: Yes

4. Is the manuscript presented in an intelligible fashion and written in standard English?

Reviewer #1: Yes

Reviewer #2: Yes

Reviewer #1: In this study, the authors propose a novel method for identifying influential nodes that are effective for diffusion in complex networks. This method introduces the concept of exclusive neighboring nodes in addition to conventional neighboring nodes. Numerical experiments demonstrate that this approach can identify nodes that facilitate diffusion more effectively than existing methods.

The introduction of exclusive neighborhoods presents a fresh and intriguing perspective. The effectiveness of this concept has also been validated through numerical experiments.

However, several aspects of the study are not fully explained, which necessitates a major revision.

Major Comments:

1. Clarification of logarithm usage in Equations (4) and (5)

Please provide a detailed explanation of why logarithms are used in these equations. At a minimum, state the advantages of this approach and the reason behind its implementation.

2. Interpretation of the final formula (Equation 9)

The study introduces "exclusive neighboring nodes" as a new perspective for identifying influencers. However, the calculation of this index is quite complex, making it unclear what the final metric truly represents. While specific calculation examples are provided (lines 274–302), it would be more informative to illustrate the network structure of high-ranking influencers based on this metric. In other words, the paper should explicitly clarify which nodes are identified as influencers and how their evaluation is influenced by the introduction of exclusive neighborhoods.

3. Clarification of red and blue numbers in the tables

I could not find an explanation for the meaning of the red and blue numbers in Tables 1, 3, 4, and 5. Please specify what these color distinctions represent.

Minor Comments:

4. Notation consistency in Equation (5)

The term EN(v_j) should be written as EN(v_i, v_j) to maintain consistency with the notation used in Equation (3).

5. Notation clarification in line 285

The term max ND appears to refer to max k_{max}, but the notation should be aligned for consistency throughout the manuscript.

Reviewer #2: This is a really interesting method and a fantastic introduction to the general subject matter. I really enjoyed the introduction and background sections. The method is clearly described; although I think it could be made more concise without losing clarity, I prefer the longer explanation. Small comment there is that the notation "Ks value" for "k-shell" was never introduced.

My main concern with this work is the level of claims for every result section that this method is "superior", "more consistent", etc. I think a little bit more humility would still leave the method looking great (it seems like a great method!), while acknowledging that other methods out-perform it at times. It was a bit tedious to read an enthusiastic claim at the end of every results paragraph. Highlighting some areas where other models out-perform MNEN would seem appropriate, and generally listing comparison metrics for the other models would seem like better use of space than simply listing all the top nodes for each model.

**Do you want your identity to be public for this peer review?** For information about this choice, including consent withdrawal, please see our Privacy Policy

Reviewer #1: **Yes: ** Masaki Chujyo

Reviewer #2: No

---

## [Author Response · Author response to Decision Letter 1]

7 Jul 2025

Dear reviewers:

We are very grateful to your consideration and insightful comments for our manuscript # PONE-D-25-05440 entitled “Influential node identification method based on multi-order neighbors and exclusive neighborhood”. Thank you very much for the constructive suggestions and for providing us with an opportunity to improve it. In the following we give point-by-point responses to your comments. All responses to reviewers and corrections to the manuscript have been marked in blue. For a detailed account of the changes, please refer to the "Response to Reviewers" document.

---

## [Decision Letter · Decision Letter 1]

29 Jul 2025

Influential node identification method based on multi-order neighbors and exclusive neighborhood

PONE-D-25-05440R1

Dear Dr. Sun,

We’re pleased to inform you that your manuscript has been judged scientifically suitable for publication and will be formally accepted for publication once it meets all outstanding technical requirements.

Kind regards,

Giridhar Maji, Ph.D.

Academic Editor

PLOS ONE

Additional Editor Comments (optional):

Reviewers' comments:

Reviewer's Responses to Questions

**Comments to the Author**

Reviewer #1: All comments have been addressed

2. Is the manuscript technically sound, and do the data support the conclusions?

Reviewer #1: Yes

3. Has the statistical analysis been performed appropriately and rigorously?

Reviewer #1: Yes

4. Have the authors made all data underlying the findings in their manuscript fully available?

Reviewer #1: Yes

5. Is the manuscript presented in an intelligible fashion and written in standard English?

Reviewer #1: Yes

Reviewer #1: The authors have responded sincerely to the reviewers' comments, and as a result, the paper has been improved. I therefore recommend acceptance.

**Do you want your identity to be public for this peer review?** For information about this choice, including consent withdrawal, please see our Privacy Policy

Reviewer #1: No

---

## [Editor Report · Acceptance letter]

PONE-D-25-05440R1

PLOS ONE

Dear Dr. Sun,

I'm pleased to inform you that your manuscript has been deemed suitable for publication in PLOS ONE. Congratulations! Your manuscript is now being handed over to our production team.

Kind regards,

on behalf of

Dr. Giridhar Maji

Academic Editor

PLOS ONE